# Bim Expression Modulates Branching Morphogenesis of the Epithelium and Endothelium

**DOI:** 10.3390/biom12091295

**Published:** 2022-09-14

**Authors:** Christine M. Sorenson, Yong-Seok Song, Shoujian Wang, Soesiawati R. Darjatmoko, Mohammad Ali Saghiri, Mahsa Ranji, Nader Sheibani

**Affiliations:** 1Department of Pediatrics, University of Wisconsin School of Medicine and Public Health, Madison, WI 53792, USA; 2McPherson Eye Research Institute, University of Wisconsin School of Medicine and Public Health, Madison, WI 53705, USA; 3Department of Ophthalmology and Visual Sciences, University of Wisconsin School of Medicine and Public Health, Madison, WI 53705, USA; 4Biomaterial and Prosthodontic Laboratory, Department of Restorative Dentistry, Rutgers School of Dental Medicine, Newark, NJ 07103, USA; 5EECS Department, I-Sense and I-Brain, Florida Atlantic University, Boca Raton, FL 33431, USA; 6Department of Cell and Regenerative Biology, University of Wisconsin School of Medicine and Public Health, Madison, WI 53705, USA; 7Department of Biomedical Engineering, University of Wisconsin School of Medicine and Public Health, Madison, WI 53706, USA

**Keywords:** kidney development, ureteric bud, retinal vasculature

## Abstract

Branching morphogenesis is a key developmental process during organogenesis, such that its disruption frequently leads to long-term consequences. The kidney and eye share many etiologies, perhaps, due to similar use of developmental branching morphogenesis and signaling pathways including cell death. Tipping the apoptotic balance towards apoptosis imparts a ureteric bud and retinal vascular branching phenotype similar to one that occurs in papillorenal syndrome. Here, to compare ureteric bud and retinal vascular branching in the context of decreased apoptosis, we investigated the impact of Bim, Bcl-2’s rival force. In the metanephros, lack of Bim expression enhanced ureteric bud branching with increases in ureteric bud length, branch points, and branch end points. Unfortunately, enhanced ureteric bud branching also came with increased branching defects and other undesirable consequences. Although we did see increased nephron number and renal mass, we observed glomeruli collapse. Retinal vascular branching in the absence of Bim expression had similarities with the ureteric bud including increased vascular length, branching length, segment length, and branching interval. Thus, our studies emphasize the impact appropriate Bim expression has on the overall length and branching in both the ureteric bud and retinal vasculature.

## 1. Introduction

The eye has been described as a window to the kidney. Thus, it is not surprising that the kidney and eye share many signaling pathways during development. Perhaps as a consequence many diseases including Alport, Senior Lɸken, and papillorenal syndromes share kidney and eye etiologies. However, the identity of these shared pathways and consequences of their dysregulation needs further investigation.

Modulation of apoptosis plays a prominent role during development and maintenance in many organs including the kidney and eye. Bcl-2 discovered due to its anti-apoptotic activity, was the first family member identified in this cell death regulatory family [1,2,3]. Our previous studies demonstrated that lack of Bcl-2 expression disrupted retinal arterial and capillary branching [4,5]. These mice develop a syndrome similar to papillorenal syndrome with vessels in the central retina entering and bifurcating near the disc periphery [4,6]. Moreover, similar to papillorenal syndrome, mice globally lacking Bcl-2 expression exhibit renal hypoplasia in part due to decreased distance of the first ureteric bud branch point and subsequent reduced ureteric bud branching [7]. Thus, an inability to properly temper apoptosis is linked to deficiencies in ureteric bud and retinal vascular branching in the kidney and eye, respectively.

Bim is a pro-apoptotic Bcl-2 family member that removes unnecessary cells. It acts in opposition to Bcl-2, such that removal of a single Bim allele is sufficient to prevent renal hypoplasia/cystic dysplasia in Bcl-2 −/− mice [8,9]. Reiterative ureteric bud branching morphogenesis generates the collecting duct system. We have examined the impact Bim expression has in postnatal collecting duct and renal endothelial cells in vitro. These studies previously demonstrated that Bim −/− postnatal collecting duct cells are less migratory, more adherent and expressed higher levels of thrombospondin-1 (TSP1), osteopontin, and VEGF compared to their wild-type counterparts, while Bim −/− kidney endothelial cells had increased migration, VEGF expression, and capillary morphogenesis with decreased expression of extracellular matrix proteins TSP1 and osteopontin compared to wild-type counterparts [10]. In vivo, these changes equated to a significant increase in the numbers of peritubular capillaries in kidneys from postnatal day 28 (P28) Bim −/− mice [10]. However, much remains to be explored regarding the role Bim plays during nephrogenesis and how it mirrors what occurs in the eye.

In the eye, Bim plays a pivotal role in hyaloid vessel regression, hyperoxia-mediated retinal vessels obliteration, and developmental retinal vascular pruning [11]. Bim −/− postnatal retinal endothelial cells and pericytes demonstrated increased migration, adhesion, and VEGF expression in vitro [12]. Although these data indicated a role for Bim expression during development, little is known regarding whether aberrant branching occurs in the ureteric bud and retinal vasculature in the absence of Bim expression in vivo. Gaining a better understanding of the role Bim plays in branching morphogenesis will give us better insight into renal and ocular morphogenesis when dysregulated apoptosis predominates.

Here we assessed the impact lack of Bim expression has on formation of a branched network in the ureteric bud and retinal vasculature. In the metanephric kidney, lack of Bim expression impacted ureteric bud branching morphogenesis. The ureteric bud T-shape branch originated higher and was skewed from the midline in the absence of Bim. Unfortunately, enhanced ureteric bud branching came with increased branching defects including single or multiple varying sized pieces of ureteric bud which was not attached to the intact branching ureteric bud as well as bifurcation of the ureteric bud once it entered the confines of the metanephric mesenchyme.

The impact of enhanced ureteric bud branching in Bim −/− mice lingered in the postnatal kidney. At birth, a modest but significant increase in renal mass and nephrogenic zone size was seen in Bim −/− mice. Later, at postnatal day 21 (P21), we observed increased renal mass and glomeruli numbers accompanied by decreased oxidative stress. We also noted decreased numbers of podocytes per glomerulus with some glomeruli from Bim −/− mice demonstrating collapse. In the absence of Bim expression, retinal vasculature branching revealed branching similarities with ureteric bud including increased total length, branching length, and branching interval in Bim −/− mice. Thus, lack of Bim expression increases organ complexity through excess branching. In the future this insight will give us the potential to enhance or maintain nephron number or capillary density to improve organ function and/or vision.

## 2. Methods

### 2.1. Mice and Metanephric Organ Culture

Bim +/− breeder pairs (Stock number 004525; Jackson Laboratory, Bar Harbor, ME, USA) maintenance and studies were approved by the institutional animal care and use committee of the University of Wisconsin School of Medicine. Genotyping of mice and embryos was accomplished by PCR of extracted genomic DNA using the following primers 5′-CATTCTCGTAAGTCCGAGTCT-3′, 5′-GTGCTAACTGAAACCAGATTAG-3′, and 5′-CTCAGTCCATTCATCAACAG-3′ [11].

We surgically dissected kidneys from embryonic day 12 embryos (E12; T-shaped ureteric bud) and placed them on a Transwell clear polyester membrane (0.4 μm; Costar 3460, Corning, NY, USA). All embryos were genotyped as noted above. Organ cultures were maintained with DMEM: F12 medium (Thermo Fisher Scientific, Carlsbad, CA, USA) supplemented with 5XMITO (BD Biosciences, Franklin Lakes, NJ, USA), 50 units/mL penicillin, 50 μg/mL streptomycin (Sigma, St Louis, MO, USA), 50 μg/mL gentamicin (Thermo Fisher Scientific, Carlsbad, CA, USA), and 50 unit/mL nystatin (Sigma, St Louis, MO, USA), grown for four days at 37 °C, and photographed with a Nikon microscope. Next, to visualize the ureteric bud, the embryonic kidneys were wholemount stained with anti-cytokeratin [13]. The kidneys were fixed in cold methanol, permeabilized with 0.1% triton X100 and incubated overnight in anti-cytokeratin (1:100; C2562, Sigma, St Louis, MO, USA) in PBS/tween 0.1% with 5% normal goat serum (NGS) while shaking as previously described [13]. The next day the kidneys underwent three exchanges of PBS with 0.1% Tween and the appropriate secondary (715-165-151; Jackson ImmunoResearch, West Grove, PA, USA) was added overnight while shaking. On the subsequent day, there were, again, three washes and then mounting in PBS/glycerol (1:1) with Grace Bio-labs coverslip (Bend, OR, USA. PC 20). We quantified branching in the ureteric bud using the Angiogenesis Analyzer plug-in for ImageJ to analyze cellular branched networks.

### 2.2. Vascular Staining of Wholemount Kidneys

Kidneys from P0 mice were fixed in 4% paraformaldehyde, then moved to cold 100% methanol, and stored at −20 °C until use. For anti-α-smooth muscle actin (SMA) staining kidneys were washed with PBS, blocked (50% FCS/20%NGS/0.01% tritonX-100 in PBS), and incubated with anti-SMA-FITC (1:250; F3777; Sigma, in blocking solution) overnight at 4 °C. The kidneys were washed, mounted in PBS: glycerol and cover slipped with Grace Bio-Labs coverslip.

### 2.3. Processing of Kidneys for Histological Studies and Immunochemistry

Kidneys from P0 and P21 mice were fixed overnight in formalin and processed for paraffin sectioning. Paraffin sections were deparaffinized with xylene and rehydrated and were stained with hematoxylin and eosin (H&E). To unmask the antigen we used antigen-unmasking solution (H-3300; Vector Laboratories, Burlingame, CA, USA), the sections were washed in phosphate buffered saline (PBS) and incubated in blocking buffer (PBS containing 1% bovine serum albumin, 0.3% Triton X-100, and 0.2% skim milk powder) containing anti-Ki-67 (1:100; TEC-3; DAKO Glostrup, Denmark), anti-Pax-2 (1:100; #SC-7747 Santa Cruz, CA, USA), or anti-WT-1 (1:100; SC-192. In some cases, these slides were double stained with *Dolichos biflorus* agglutinin or *Lycopersicon esculentum* (1:40; FL1031 or FL1321, Vector Laboratories). The sections were then incubated with the appropriate secondary indocarbocyanine (CY3)-labeled antibody (Jackson ImmunoResearch, West Grove, PA, USA). The slides were imaged with a Nikon microscope equipped with a digital camera. The depth of the nephrogenic zone was determined by measuring in six areas around the kidney periphery of a P0 mouse. Nephrogenic zones from six mice of each genotype were measured, and a mean was obtained. Glomeruli per midsagittal were determined by counting total glomeruli numbers on a midsagittal kidney section from P21 mice (six mice each genotype), and a mean was determined. Podocyte numbers were assessed by counting WT-1 staining cells per glomerulus on a mid-sagittal kidney section of P21 mice (six mice each genotype), and a mean was determined.

### 2.4. Cryo Fluorescence Redox Studies

Mitochondrial metabolic coenzymes NADH (nicotinamide adenine dinucleotide), and FADH2 (flavoprotein adenine dinucleotide) are the primary electron carriers in oxidative phosphorylation. Since NADH and FAD (oxidized form of FADH2) are autofluorescent they can be monitored without exogenous labels by noninvasive optical techniques [14]. The 3D cryoimager utilizes an excitation light source of a 200 W mercury arc lamp that is filtered at the excitation wavelengths of NADH and FAD. The excitation band pass filters used for NADH was 350 nm (80 nm bandwidth, UV Pass Blacklite, HD Dichroic, Los Angeles, CA) and for FAD was 437 nm (20 nm bandwidth, 440QV21, Omega Optical, Brattleboro, VT), and the emission filters for NADH was 460 nm (50 nm bandwidth, D460/50M, Chroma, Bellows Falls, VT) and for FAD was 537 nm (50 nm bandwidth, QMAX EM 510–560, Omega Optical, Brattleboro, VT), as we previously described in detail. For this study, we used a resolution of 10 μm in the z direction (~500 z-slices per kidney) in cryo-sectioning. We used MATLAB (The MathWorks, Inc., Natick, MA, USA) to process FAD and NADH autofluorescence images (500 slices per kidney) and composite images were created with both NADH and FAD signals. The NADH to FAD ratio (mitochondrial redox ratio, was calculated voxel by voxel, using MATLAB, according to Redox Ratio = RR = NADH/ FAD.

### 2.5. Fundus Imaging

Bim +/+ and Bim −/− (ten-week-old) mice were anesthetized (ketamine 100 mg/kg and xylazine 10 mg/kg). Their pupils were dilated with 1% tropicamide (Bausch and Lomb Inc., Tampa, FL, USA) and the mice kept warm on a heating pad during the experiment. A Micron III retinal imaging system (Phoenix Laboratories Inc., Pleasanton, CA, USA) was used to perform fundus imaging on both eyes from all mice examined. We used an equal image depth, the same magnification, and made sure that the circular border of the image was distinctly in focus. The images were then analyzed as previously described [15] using Angiogenesis Analyzer plug-in for ImageJ to analyze cellular networks.

### 2.6. Statistical Analysis

Statistical differences between two groups were evaluated with the student’s unpaired t-test (two-tailed). We evaluated statistical differences between groups with one-way ANOVA followed by Tukey’s multiple comparison test using GraphPad Prism 8.0 (GraphPad Software, San Diego, CA, USA). The mean ± SEM was shown. *p* < 0.05 was considered significant.

## 3. Results

### 3.1. Bim Expression Modulates Ureteric Bud Branching

Bim is expressed in the developing kidney in the mesenchyme of the metanephric cap at E13.5-14.5 [8]. This is the region of the metanephros that undergoes fulminant apoptosis in the absence of Bcl-2 [16]. To assess the impact that Bim expression has on ureteric bud branching we harvested embryonic day 12 (E12) kidneys and placed them in organ culture for three days. At day three, phase images were captured and the metanephroi stained with anti-cytokeratin to visualize the ureteric bud (Figure 1A). As demonstrated in Figure 1A,C, Bim −/− embryos had increased ureteric bud branch points, end branch tips and total ureteric bud length compared to Bim +/+ littermates. Thus, the lack of Bim expression increased ureteric bud branching.

Our previous studies on mice lacking Bcl-2 demonstrated a shortened distance to the first ureteric bud branch point (T-shaped branch) following entry to the mesenchyme [7]. Since Bim acts in opposition toward Bcl-2 here we assessed whether Bim expression influenced ureteric bud branching near the initial T-shaped branch point. We assessed the distance from the point ureteric bud enters the mesenchyme to its initial T-shaped branch, distance of T-shaped branch to midline and the distance to the subsequent branch point (Figure 1B,C). Metanephroi from Bim −/− embryos demonstrated disrupted branching symmetry due to a significant increase in the distance from the point ureteric bud enters the mesenchyme to its initial T-shape and significant skewing of the ureteric bud from midline (Figure 1). Secondary branching of the ureteric bud, in close proximity to the T-shape branch (<125 µm), was also significantly increased in the absence of Bim. We also noted that ~26% of metanephroi from Bim deficient embryos had single or multiple varying sized pieces of ureteric bud that did not appear to be attached to the main branching ureteric bud and in nearly 8% of kidneys the ureteric bud separated or bifurcated within the confines of the metanephric mesenchyme (**** *p* < 0.0001 and **p* < 0.05, respectively, compared to Bim +/+ littermates). Thus, although ureteric bud branching is enhanced in the absence of Bim, the initial T-shaped branch is higher into the mesenchyme and skewed from the midline.

### 3.2. Increased Renal Mass and Nephrogenic Zone Depth in the Absence of Bim

Due to the aberrant ureteric bud branching noted in the absence of Bim, we next assessed whether these changes impacted the kidney postnatally. Pax2 is expressed in induced mesenchyme, and is essential for epithelial conversion [17]. In newborn kidneys, Pax2 staining is mainly confined to the nephrogenic zone, where we also observed the highest level of proliferation (Ki-67 staining). We observed overlapped Pax2 staining and proliferation in the nephrogenic zone of Bim +/+ and Bim −/− mice. The nephrogenic zone depth was increased and had a less compressed appearance in Bim −/− mice compared to wild-type mice (Table 1; Figure 2 and Figure 3). At birth, kidneys from Bim −/− mice had a modest increase in renal mass (Table 1). Later at P21, renal mass increased significantly in Bim −/− mice compared to their Bim +/+ counterparts, which corresponded to a nearly two-fold increase in the mid-sagittal glomerular numbers (Table 1). However, glomeruli from Bim −/− mice had lower podocyte numbers per glomeruli (Table 1). We also noted that glomeruli in Bim −/− mice demonstrated collapse (Figure 4). Thus, although kidneys from Bim −/− mice have increased renal mass with increased glomeruli numbers, some glomeruli underwent collapse.

### 3.3. Decreased Oxidative Stress in the Absence of Bim

Our previous studies demonstrated increased oxidative stress in kidneys from Bcl-2 −/− mice. Since Bim typically acts in opposition to Bcl-2, here we asked whether the mitochondrial redox ratio was affected in the absence of Bim expression. We have demonstrated that the normalized ratio of intrinsic mitochondrial fluorophores (NADH/FAD, the mitochondrial redox ratio) is a marker of the mitochondrial redox and metabolic state of tissue ex vivo and in vivo [18]. In Figure 5, cryo-fluorescence redox imaging shows kidneys from Bim −/− mice had decreased oxidative stress (higher redox ratio). This is consistent with Bim promoting apoptosis by negating the antioxidant effect of Bcl-2.

### 3.4. Increased Retinal Capillary Length and Branching Interval in the Absence of Bim Expression

The studies presented here demonstrated enhanced ureteric bud branching in the absence of Bim. To examine the impact Bim expression has on retinal vascular branching we obtained fundus images from Bim +/+ and Bim −/− mice. Utilizing, together, the Canny Edge Detector and Angiogenesis Analyzer plug-in, we analyzed retinal microvascular branching characteristics to assess total length (sum of all length for all segments, isolated segment, and branches in analyzed area), branching length (combination of all segment and branch length in the area), total segment length (length of extremities with node on both ends), branching interval (mean distance separating two branches), number of master junctions (junctions connecting at least two master segments), number of branches (extremities connected to vasculature by only a single junction), and number of extremities (any and all unique pieces of the vasculature). We also assessed whether branching or tortuous vessels were visual on the superficial layer readily observed on the fundus image. We have successfully utilized this technique previously to detect early temporal retinal vascular changes with diabetes [15]. This strategy allowed us to quantitate the quality of the retinal vascular architecture by identifying vascular-based abnormalities between Bim +/+ and Bim −/− mice. Here we showed that retinas from Bim −/− mice display increased total length, branching length, segment length, and branching interval compared to Bim +/+ littermates. The number of branches and extremities were similar in Bim +/+ and Bim −/− mice, while the number of master junctions in the retinal vasculature decreased in the absence of Bim (Figure 6). In the most superficial layer, we also noted increased numbers of vessels with branched and tortuous vessels in the absence of Bim (Figure 6). Thus, similar to the ureteric bud, the overall length of the retinal vasculature increased as well as many branching indices.

### 3.5. Increased Renal Capillary Loops Prior to Birth in Kidneys from Bim −/− Mice

Similar to the retinal vasculature where the retinal artery bifurcates near the optic nerve, the renal artery bifurcates near the hilum. Here we stained embryonic kidneys with anti-α-smooth muscle actin to visualize the vasculature. At E16, the main renal arteries are visible in both Bim +/+ and Bim −/− kidneys, although primary bifurcation of the main renal artery appears to occur farther into the kidney in Bim −/− mice (Figure 7). By E17 primary and secondary branching of the renal arteries are apparent. The complexity of the renal vasculature by E18 increases with a subcapsular vascular network appearing in both Bim +/+ and Bim −/− kidneys, although the vascular network in kidneys from Bim −/− mice appeared more complex (Figure 7).

## 4. Discussion

The process of apoptosis and the genes that facilitate it play integral roles during organ development and maturation. Apoptosis removes unnecessary cells at early stages of development, and later when branched structures, such as those in the vasculature, are remodeled. This is particularly evident in the retinal vasculature where Bim is essential for vascular pruning and remodeling [11]. Unfortunately, the contribution of renal vascular and ureteric bud pruning or remodeling and its contribution to nephron endowment to our knowledge is unknown.

Bim gene dosage regulates renal progenitor survival [19]. Our previous studies demonstrated an important role for Bcl-2 in maintaining metanephric mesenchyme viability and ureteric bud branching [7,16]. In the absence of Bcl-2, ureteric bud branch points and tips were decreased as well as the distance to the first ureteric bud branch point (T-shape). This correlated with renal hypoplasia postnatally. Bcl-2 −/− ureteric bud cells were also unable to branch in vitro [7]. Re-expression of Bcl-2 in the ureteric bud in Bcl-2 deficient mice partially rescued a normal renal phenotype [20]. Since Bim acts in opposition to Bcl-2 it is tempting to speculate ureteric bud branching and nephron endowment would be impacted in the absence of Bim. Here, Bim −/− embryos demonstrated increased ureteric bud branch points and tips indicating enhanced ureteric bud branching morphogenesis. However, enhanced ureteric bud branching came with some notable costs. Kidneys from Bim −/− embryos formed the first T-shaped branch point further into the mesenchyme and the branch point was significantly skewed from midline. We also noted that about a quarter of the kidneys from Bim −/− embryos displayed unattached ureteric bud fragments within the mesenchyme or ureteric bud that split or bifurcated upon entering the mesenchyme. Thus, Bim expression assists in choregraphing appropriate branching of the ureteric bud.

Bim is well known for its ability to promote cell death. In its absence enhanced survival of progenitor cells and diminished remodeling could contribute to aberrant enhanced numbers of branches if little to nothing is eliminated. Our data illustrates how with minor increases in branching at early stages, each round of dichotomous branching doubles the number of ureteric tips generating increased nephron numbers, as seen in the absence of Bim. Our previous studies have noted increased adhesion as well as aberrant extracellular matrix production and migration in the absence of Bim [10]. Thus, it is tempting to speculate that increased adhesion coupled with aberrant extracellular matrix production pulls the ureteric bud off its normal course causing fragments of ureteric bud to be left behind in its wake.

Apoptosis plays an important role during formation of the vascular network. Normal tissue mass/organ size is tightly regulated by the degree of vascularization [21,22]. The renal vasculature begins forming embryonically revealing extensive pattern of renal arterial branches. We previously noted decreased TSP1 expression in Bim −/− kidney endothelial cells which given our previous studies would indicate decreased vascular remodeling [12]. TSP1 is an endogenous inhibitor of angiogenesis that can limit blood vessel density in normal tissue. TSP1 deficient mice exhibit a significant increase in blood vessel density in many organs including retina [23,24], supporting the role of TSP1 in the regulation of angiogenesis in vivo. We have shown that TSP1 plays an essential role in retinal vascular pruning [25]. Decreased TSP1 expression in Bim −/− renal vascular cells could increase glomerular numbers due to attenuated vascular pruning.

Bim expression plays an important role in the eye through its regulation of ocular vascular cell survival. In the eye, vascular remodeling and regression have prominent roles during development and in pathologic states. Bim not only facilitates removal of unwanted cells as during hyaloid vessel regression [11], but is also responsible for pericyte cell death during pathologic states such as hyperglycemia driven oxidative stress with diabetes and retinal vessel loss during hyperoxia [11,26]. The role reactive oxygen species (ROS) play in controlling developmental is beginning to emerge. In the absence of Bim, FOXO3-induced ROS production is reportedly prevented [27], consistent with decreased oxidative stress noted here in kidneys from Bim −/− mice. We have also shown that increased oxidative stress in the absence of Bcl-2 disrupts endothelial branching which is restored by addition of the antioxidant N-acetylcysteine [28], giving credence to the roles Bim and Bcl-2 have in modulating the impact of ROS. Retinal endothelial cell and pericyte numbers also increase in the absence of Bim, although its impact on retinal vascular branching was not known [11,29]. Cotters group observed increased retinal thickness in the absence of Bim, which may be the ocular parallel to increased renal mass corresponding to increased vascular density [30]. Bim expression is also important for optic nerve head development which may along with vascular density directly influence glaucoma pathophysiology by its ability to impact neurodegeneration [30,31]. Thus, the ability of Bim expression to modulate vascular programming is pivotal in the eye.

In summary, given overlapping pathologies in the kidney and eye as well as developmental programming, we assessed the similarities of branching in the ureteric bud and retinal vasculature that occurred in the absence of Bim. Our previous studies demonstrated similar unique branching characteristics in both the ureteric bud and retinal vasculature in the absence of Bcl-2 [4,5,7]. Here we demonstrated several similarities with overall ureteric bud and retinal vascular length significantly increasing in the absence of Bim. In the retinal vasculature, we noted increased branch length and branching interval. In the kidney this correlated with and increased distance to the first ureteric bud branch point (T-shape). Thus, Bim plays important roles during branching morphogenesis affecting tissue developmental processes and function.

## Figures and Tables

**Figure 1 biomolecules-12-01295-f001:**
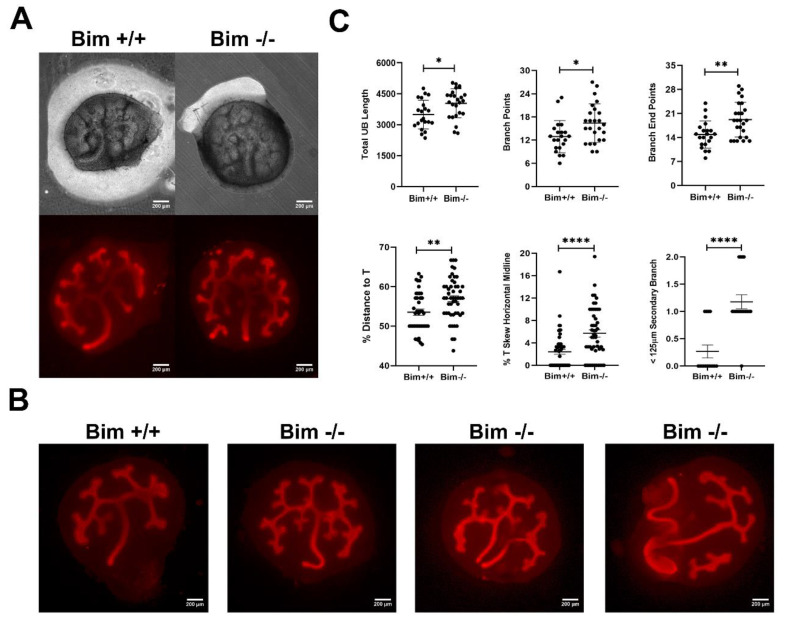
Enhanced ureteric bud branching in the absence of Bim. In Panel (**A**), phase images were taken of embryonic kidneys from E12 Bim +/+ and Bim −/− mice grown in culture for three days (upper panel), and then wholemount stained with anti-cytokeratin to visualize the ureteric bud (lower panel). Panel (**B**) are more representative images demonstrating some of the variation in ureteric bud branching noted in the absence of Bim. Panel (**C**) is the quantitation of total ureteric bud length, branch points, branch end points, percent distance to the T-branch, percent skew from midline of the T-branch, and number of secondary branches the occurred closer than 125 µm. Data are representative of 25 animals. Scale bar is 200 µm. * *p* < 0.05, ** *p* < 0.01, and **** *p* < 0.0001.

**Figure 2 biomolecules-12-01295-f002:**
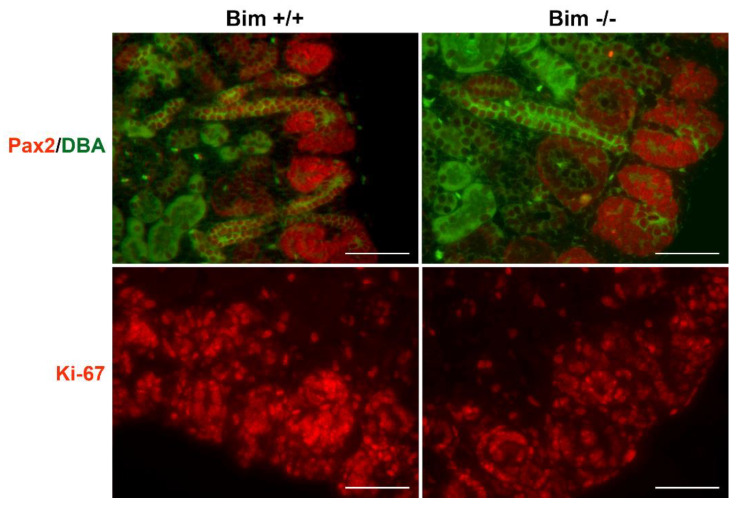
Nephrogenic zone appears less compressed in kidneys from Bim −/− mice. Kidneys from P0 mice were stained with anti-Pax2 (red) and DBA (green) in the upper panel. In the lower panel Ki-67 staining was conducted to identify proliferating cells. Data representative of six mice. Scale bar is 12.5 µm.

**Figure 3 biomolecules-12-01295-f003:**
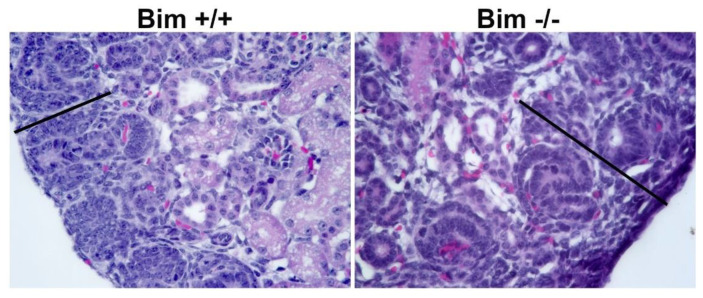
Increased nephrogenic zone depth Bim −/− mice. Nephrogenic zone depth was calculated by measuring six nephrogenic zone areas (note line) on the periphery of each kidney from P0 Bim +/+ and Bim −/− mice and is summarized in Table 1. Data representative of six mice. Magnification is 400×.

**Figure 4 biomolecules-12-01295-f004:**
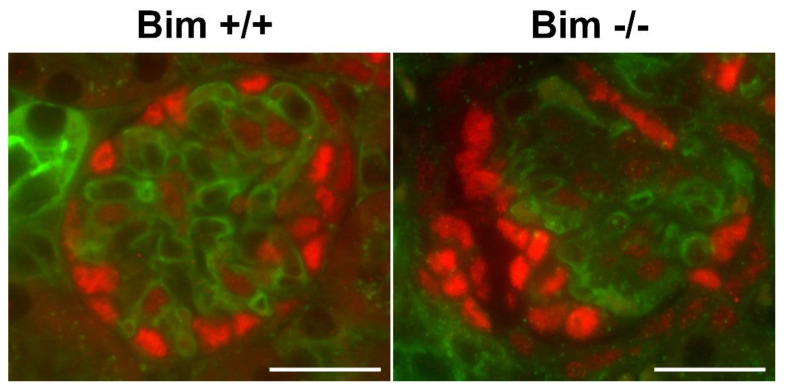
Glomular collapse in kidneys from P21 Bim −/− mice. Kidneys from P21 mice were stained with WT-1 (red) to stain podocytes and *Lycopersicon esculentum* (green) to stain the vasculature. Data is representative of six mice. Scale bar is 100 µm.

**Figure 5 biomolecules-12-01295-f005:**
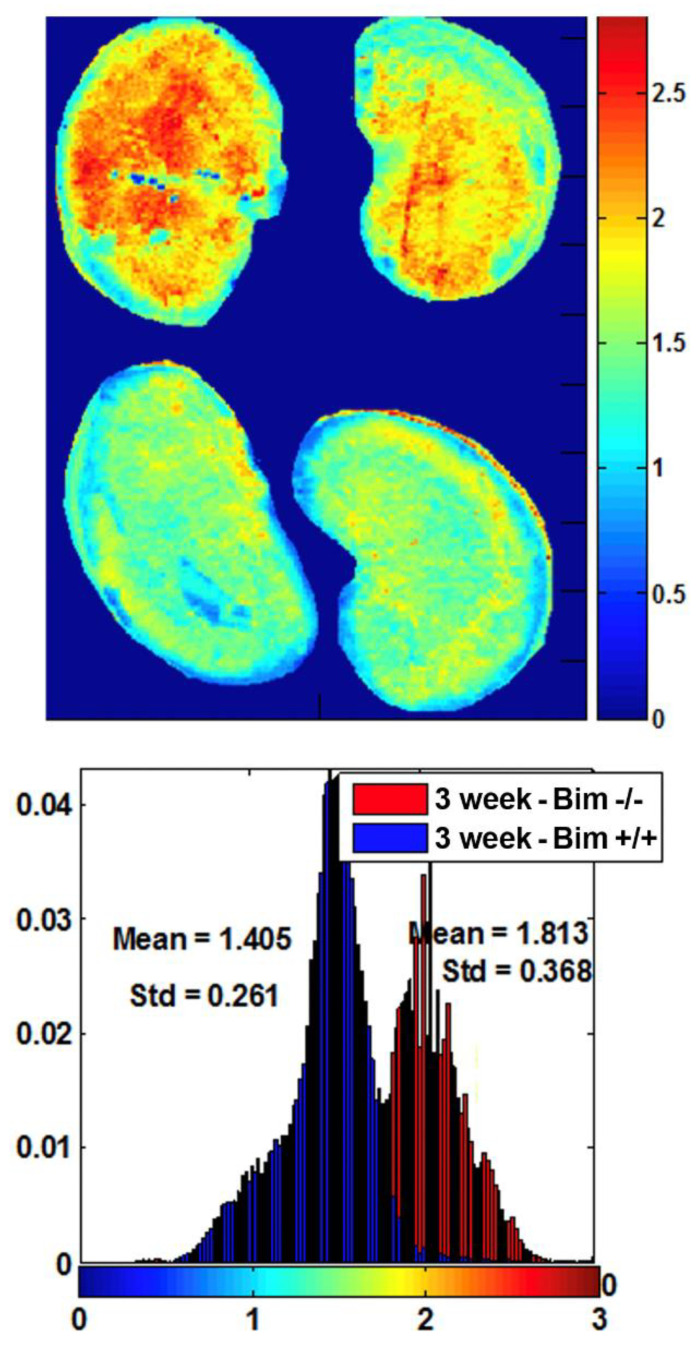
Decreased oxidative stress in kidneys from Bim −/− mice. Top panel: Low resolution NADH redox ratio image of kidneys from Bim −/− (top row) wild-type control mice (bottom row). Bottom panel are histograms of the redox ratio distribution in each group. Please note decreased oxidative stress in the absence of Bim. Data representative of six mice.

**Figure 6 biomolecules-12-01295-f006:**
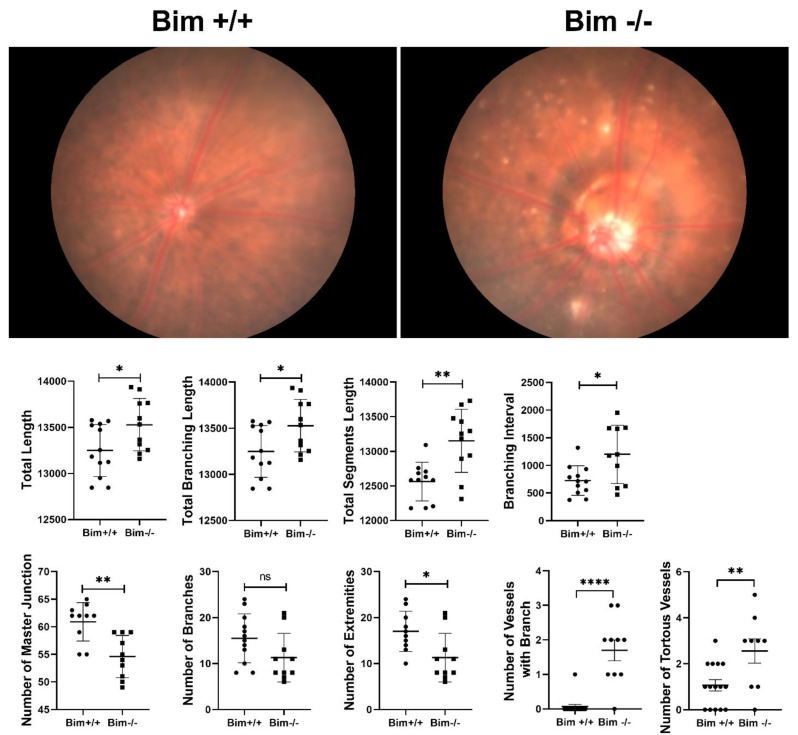
Increased retinal branching in the absence of Bim. Fundus imaging was performed on ten week-old Bim +/+ and Bim −/− mice utilizing a Micron III imaging system. The Canny Edge Detector and Angiogenesis Analyzer plug-in were used to analyze retinal microvascular branching characteristics. We also assessed branching or tortuous vessels visual on the superficial layer on the fundus image. Data representative of six mice. * *p* < 0.05, ** *p* < 0.01, and **** *p* < 0.0001, ns = not significant.

**Figure 7 biomolecules-12-01295-f007:**
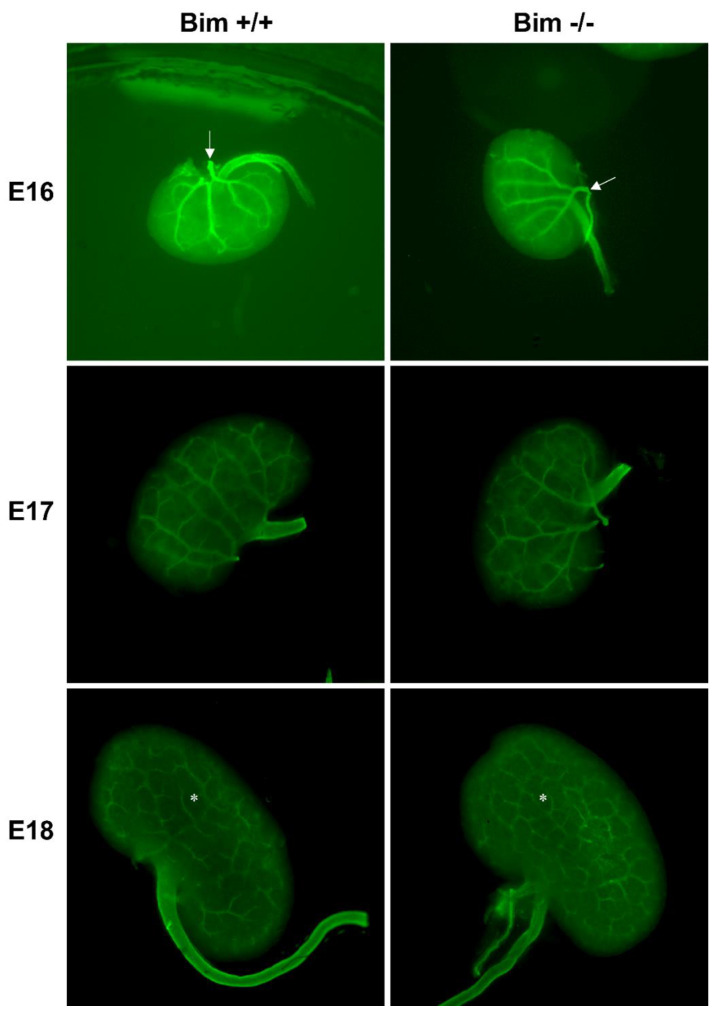
Increased capillary loops at E18 in Bim −/− mice. Kidneys from E16–18 embryos were wholemount stained for α-SMA to visualize the renal vasculature. Images are taken at 20× magnification. Please note Bim +/+ and Bim −/− mice are littermates. At E16 arrow points to primary bifurcation of the renal artery. At E18 the asterisk denotes subcapsular vascular network. Data representative of six mice.

**Table 1 biomolecules-12-01295-t001:** Kidneys from Bim −/− mice have increased mass. Kidney weight (mg), mid-sagittal glomerular and podocyte numbers for postnatal day 21 (P21) mice, kidney weight, and nephrogenic zone depth (see Figure 3) are summarized for P0 mice. * *p* < 0.05, ** *p* < 0.01, and **** *p* < 0.0001.

**P0**	**Kidney** **Weight (mg)**	**Nephrogenic** **Zone Depth (mm)**	
Bim +/+	7.7 ± 1.3	0.14 ± 0.01	
Bim −/−	8.8 ± 1 *	0.19 ± 0.01 *	
**P21**	**Kidney Weight (mg)**	**Glomeruli/midsagittal**	**Podocyte number/Glomerulus**
Bim +/+	69 ± 6	125 ± 7	13.7 ± 2
Bim −/−	94 ± 9 ****	245 ± 12 ****	9.5 ± 2 **

## Data Availability

Not applicable.

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
