# Peer review of "Bim Expression Modulates Branching Morphogenesis of the Epithelium and Endothelium"

_biomolecules, 2022, doi:10.3390/biom12091295_

Round 1
Reviewer 1 Report
Prior work from this group has demonstrated that Bcl-2 knockout mice have decreased retinal vascular branching and renal hypoplasia. In this paper, the authors ask if the lack of Bim, a pro-apoptotic factor that acts in opposition to Bcl-2, alters vascular and ureteric bud branching. This is an important and interesting question because ureteric bud branching affects the total number of nephrons that will form. Very little is known about developmental pruning of renal vasculature to form the adult vascular network.
For this study, the authors examined used a Bim KO mouse model (BCL2l11tm1.1Ast). To examine UB branching, they measured branching parameters in organ-cultured WT and Bim -/- kidneys. Additionally, they measured kidney weight and other features in new-born and weanling mice. These are straight forward and appropriate methods to answer if Bim -/- mice have altered UB branching.
The current manuscript needs additional editing to clarify how the results relate to their hypothesis. In particular, the oxidative stress and retinal vascular patterning seemed out of place.
Specific points:
Figure 1/section 3.1. A diagram of an organ-cultured kidney with the various measurements clearly marked would greatly improve this figure. At E12, there is variation in developmental stages within mice from the same dam. How do you control for this variability? It would be useful to include images from all the mice used in Figure 1B in a supplement. In Figure 1C, I am not sure how to interpret the right two Bim-/- kidneys. Were the kidneys damaged during dissection? How often do you see double ureters in the third panel of figure 1C? Have you looked at E10.5 Bim -/- embryos to see if there are multiple ureteric buds growing into the metanephric mesenchyme?
Is the branching asymmetry seen in organ-cultured kidneys replicated in vivo? Are the P21 or adult kidneys misshaped? Do you see multiple ureters or renal pelvis abnormalities in the adult Bim -/- kidneys?
The last sentence of 3.2 states "although kidneys from Bim -/- mice have increased renal mass with increased number glomeruli numbers, it comes at a cost."
What is the cost? Is there any change in BUN or other renal lab values that suggests that there is altered renal function?
Table 1. Adding H&E images with notations showing how the nephrogenic depth was measured would improve this figure.
Figure 3. Because this mouse line has previously been shown to have alterations in leukocytes ( Bouillet P, Metcalf D, Huang DC, Tarlinton DM, Kay TW, Kontgen F, Adams JM, Strasser A, Proapoptotic Bcl-2 relative Bim required for certain apoptotic responses, leukocyte homeostasis, and to preclude autoimmunity. Science. 1999 Nov 26;286(5445):1735-8), the glomerular vascular changes could be immune related. How did you distinguish pseudocresent vs cresent formation in these mice? Is there any evidence for immune mediated glomerular changes in the Bim KO mice? What percent of the Bim -/- glomeruli exhibited pseudocresent formation? Did you use markers to exclude parietal epithelial cell proliferation? It would be useful to double label with WT-1 and Ki67 to evaluate if the podocytes are proliferating. Is there matrix deposition within the glomerulus?
Figure 4. The decreased oxidative stress in Bim KO kidneys is interesting, but it seems out of place in this manuscript. How does this fit in with your goal to show defects in branching morphogenesis in Bim KO mice?
Figure 5/section 3.5 . Needs more explanation about why you are switching to the retina to examine changes in vascular branching.
Discussion/Supplemental figure: It is not immediately clear that there are major differences between Bim KO and WT in the renal vasculature. This figure needs annotation to clearly identify changes in the vasculature. And it needs to be a figure, not a supplement.
Minor:At the end of section 3.1, the authors state "Thus, although ureteric bud branching is enhanced in the absence of Bim, it proceeds in its own unique way. " The phrase "its own way" should be omited.
The sentence “We wholemount SMA stained kidneys from E16-18 Bim +/+ and Bim -/- mice (Supplementary Figure 1)…” need to be modified.
Author Response
We are submitting a revised manuscript titled “Bim expression modulates branching morphogenesis of the epithelium and endothelium”. We believe the revisions the reviewer suggested have strengthened the manuscript and that it fits well with the emphasis of your special issue “Molecular mechanisms of kidney development”. The reviewer’s responses are delineated below:
Reviewer 1
- As noted by the reviewer, there is variation to the degree of development with location along the uterine horn. This is the reason we chose to dissect metanephroi from 25 animals of each genotype for these experiments to minimize this issue. I personally did all the metanephroi dissections and staining. I have 20+ years’ experience with these techniques and was careful to always assess the metanephroi integrity when dissecting. In addition, due to the dysmorphogenesis of the ureteric bud in the Bim -/- mice I was extra careful to make sure no abnormalities existed in the ureter before dissecting and the metanephroi were completely intact. Thus, I feel confident saying these metanephroi were not damaged during dissection. In further support of this conclusion we did not see similar abnormalities in the metanephroi from Bim +/+ mice. I feel that if I was inept in my dissections both genotypes should have been affected equally. In addition, I have never seen such abnormalities in any of the dissections I had done previously. In Figure 1C (now 1B) we saw the ureteric bud splitting within the confines of the mesenchyme in 4 of the 25 Bim -/- embryos we assessed. I carefully examined the UB prior to entry into the mesenchyme when dissecting and never observed multiple ureteric buds growing into the mesenchyme during dissection which is why we have not looked at a younger age such as E10.5 as the reviewer indicated.
- We have not seen any renal abnormalities in the postnatal kidney that could be attributed to the branching abnormalities such as misshapen or renal pelvis abnormalities as noted by the reviewer. Given the initial description of these mice by Bouilet etal (Science 286:1735, 1999) the ratio of Bim +/+ vs Bim -/- live births is 2:1, similar to what we observe, it is likely that embryos with significant renal branching abnormalities are in the portion of those lost prior to birth.
- The cost we alluded to with increased renal mass and nephron numbers was that of collapsing glomeruli which is now more appropriately explained in the text. Thank you for pointing out the need for better explanation. Also, as requested by the reviewer we have now included an H&E image from postnatal day 0 mouse kidneys to illustrate the nephrogenic zone measurements. Typically, the Ki-67 staining also gives a good line of demarcation for such measurements.
- As indicated by the reviewer, Bim -/- mice have compromised immune cell clearance which could lead to crescent formation. The experiments proposed by the reviewer are comprehensive and will definitively demonstrate crescent formation in these mice. Unfortunately, this is beyond to scope of this manuscript so instead we propose to change the terminology to collapsing glomeruli given we have not demonstrated an immune component here.
- We also found it interesting that kidneys from Bim -/- mice had decreased oxidative stress given in the absence of Bcl-2 we observed increased oxidative stress and these kidneys also underwent decreased ureteric bud branching. Given we previously have shown that the antioxidant N-acetylcysteine restores branching of Bcl-2 -/- retinal endothelial cells and aorta sprouting we believe the oxidative stress data here is worthy of a report in this manuscript.
- We have now better explained the bridge between ureteric bud branching and retinal vascular branching. Our previous studies examined the role Bcl-2 plays in kidney development and ureteric bud branching. We noted similar branching pattern changes in the retinal vasculature that we saw in the ureteric bud with regard to early premature branching. Although fundamentally these processes may share similar changes in branching patterns, in different organs that undergo branching morphogenesis, when lacking a functional protein most likely the players involved are tissue specific. Here we wanted to address whether lack of Bim also had a similar pattern change in branching morphogenesis in the retinal vasculature.
- The supplementary figure has now been moved to the results section as suggested by the reviewer and the points of interest noted on the figure.
- As requested “in its own way” removed.
- SMA sentence was modified as suggested. The SMA kidney staining is now incorporated into the results section as requested by the reviewer.
Reviewer 2 Report
In this manuscript, the authors characterized the phenotypic defects in the kidney and in the retina upon Bim gene knockout in the mice. The data suggest enhanced but irregular branching morphogenesis from both the kidney’s ureteric epithelium and the retina’s vasculature. The authors also reported that, at P21, Bim-/- kidneys have significantly higher weight with more glomeruli, but the podocyte number appears to be lower than that in the wild-type. Overall, there are novel aspects in this study, but it is relatively descriptive and preliminary at this stage, with no molecular mechanisms to address the observed phenomenon. In addition, some conclusions are not well supported by the evidence provided.
1. Descriptive and preliminary. Bim knockout phenotypes in the kidney described here are new, but mechanistic studies are still lacking, making the study seem descriptive and superficial.
2. Figure 1 B and C, figure legend description does not match the panel labels.
3. Pax-2 should be replaced with Pax2.
4. Figure 2, lower magnification images are needed to give an overview of the kidney section staining. Ki-67 images need a counterstain with DAPI. Ki67 should ideally co-stain with PAX2 and DBA.
5. Figure 3, the reviewer finds it difficult to see the differences between Bim+/+ and Bim-/- to be able to conclude “vascular collapse”. More description is needed to explain the criteria the authors used to define “vascular collapse” based on the images.
6. Language. It is better to describe observations clear rather than vague in scientific writing. Avoid using languages such as “…it proceeds in its own unique way…” (what unique way), and “…it comes at a cost…” (what cost).
Author Response
We are submitting a revised manuscript titled “Bim expression modulates branching morphogenesis of the epithelium and endothelium”. We believe the revisions the reviewer suggested have strengthened the manuscript and that it fits well with the emphasis of your special issue “Molecular mechanisms of kidney development”. The reviewer’s responses are delineated below:
Reviewer 2
- Thank you for your comment. We believe the changes indicated by the reviewers have greatly enhanced the manuscript.
- Thank you for pointing out the labeling of the figure. This has now been corrected.
- This has now been replaced with Pax2. Thank you for the correction.
- Thank you for your suggestion. My focus was to assess the nephrogenic zone which is why the higher power images were taken. Although having a lower magnification would give a nice overview of the kidney for Figure 2, unfortunately the slides were discarded from the cold room with a lab move and we no longer have the mice breeding.
- Thank you for the suggestion. Please see response to reviewer 1. The phrasing chosen in the original manuscript has been changed to glomerular collapse and a better description is now added to the manuscript.
- We have removed “jargon-like” phrases as suggested by the reviewer.
Reviewer 3 Report
Here, Sorenson and colleagues report the branching morphogenesis of the kidney collecting duct system (CD) and retinal capillary (RC) in Bim gene mutant mice. In the manuscript, the authors tried to claim the similarity in the mechanism of branching between CD and RC in regard to apoptosis which is regulated by Bim. However, their claim was based on wrong interpretation of previous literatures and a lack of understanding the difference in molecular mechanisms between CD and RC branchings.
First of all, the authors mentioned several syndromes to suggest the common etiologies between CD and RC branching, however, these syndromes do not always have defects in CD and RC branching morphogenesis, and more over, causative genes have been identified (eg. Col4 for Alport, Pax2 for Senior-Loken), and the lack of these genes influence to everywhere/elsewhere in kidney and retinal organogenesis more than (or other than) CD and RC branching. Hence, describing those syndromes to argue CD and RC branching is inappropriate.
Second, to investigate and compare the phenotypic changes of CD and RC branchings in Bim mutants, the author need to consider the difference of the molecular mechanisms of CD and RC branchings, especially in their tips and stalks, which are known to be regulated by totally different mechanisms. CD branching is regulated by GDNF-Ret signaling where GDNF is secreted by surrounding cap mesenchyme and its receptor Ret is expressed in tips of the ureteric tree. On the other hand, endothelial cells (EC) branching is mainly regulated by VEGF. EC stalks are able to generate a new bud in response to environmental signals and sometimes make a vascular plexus, whereas tips of CD never fused each other nor make a plexus. Without considering about this difference, it is impossible to argue about the similarities of phenotypes in CD and RC branchings in the same context of apoptosis.
Third, Bim gene is know to be involved in the regulation of apoptosis, however, in this study, there is no investigation to compare apoptosis between CD and RC.
Overall, I think this manuscript may cause a misleading to readers about "similarity of CD and RC phenotypes". The editor may consider the rejection.
Author Response
We are submitting a revised manuscript titled “Bim expression modulates branching morphogenesis of the epithelium and endothelium”. We believe the revisions the reviewer suggested have strengthened the manuscript and that it fits well with the emphasis of your special issue “Molecular mechanisms of kidney development”. The reviewer’s responses are delineated below:
Reviewer 3
Thank you for pointing out that further clarification of the correlation we are drawing needs to be improved. We are sorry for the confusion but it was not our intent to claim that the molecular mechanisms are identical between ureteric bud and retinal capillary branching. However, there may be some overlapping contributing players. As noted by the reviewer causative genes have been determined, but the influence of intersecting and/or overlapping pathways upstream and downstream of these defective genes is less understood. Thus, we feel that some of the commonalities that occur during the process of branching leaves various organs open to similar disease entities due to use of intersecting and/or overlapping pathways. Bim appears to be a key player in both systems due to the disruption seen in its absence in both organs and its expression in most organs. Because of this we feel this is an avenue to open a dialogue of such possibilities.
Here we decided to relate the etiology to branching defects to emphasize the importance of proper branching in health and disease. We are aware that the molecular mechanisms are different. However, the data presented here demonstrate Bim expression plays a significant role which is shared between the tissues discussed here.
Although Bim is typically thought of only modulating apoptosis, it impacts a number of other cell characteristics. For example, its loss causes increased VEGF expression (2-5-fold increase) which may also enhance cell survival. Its loss also results in altered ECM secretion, migration, adhesion, and proliferation that is cell and organ specific. As a further complication, the basal level of apoptosis is similar even in the absence of Bim in culture. Its only with challenge the differences in apoptosis level appear. In the whole organ during development this point of challenge is difficult to identify and given the other influences Bim has on cell matrix related functions it’s questionable whether apoptosis is the most essential function to assess.
Round 2
Reviewer 1 Report
Section 2.5 There pupils should be Their pupils
Table 1. Midsagital should be midsagittal ; podocyte number/glomerulus rather than glomeruli (assuming that you meant the average number of podocytes/each glomerulus
Figure 2. use color to denote stain color. (Pax-2 and Ki-67 in red) DBA in green. Figure is duplicated.
Section 3.3. It would be clearer if you start the paragraph with, " Our previous studies demonstrated.... Move the first sentence before "In figure 5, cryo....."
Section 3.5. Not sure what renal capillary loops refers to. Glomerular loops? Vessels surrounding the UB tips? Perhaps it would be more accurate to refer to these vessels as a subcapsular vascular network .
Discussion. Please include the oxidative stress experiment and consider shortening the TSP-1 discussion.
throughout---spell out numbers one-ten
Author Response
Thank you for pointing out the typos, phrasing changes, numbers and color label for figure 2. We also simplified the TSP1 section and added in the oxidative stress experiment in the discussion as requested. We believe these changes you suggested aided manuscript clarification and simplification.
Reviewer 2 Report
Some minor changes have been made in this revised manuscript. However, this study is still descriptive and preliminary as I commented I the first report.
Author Response
Although the reviewer feels the data is preliminary, we feel that beginning to unravel Bcl-2 pro- and anti-apoptotic family members role during branching and remodeling is novel stepping stone to new areas of research. Specifically, the unique ureteric bud branching changes we present here can be a catalyst for future investigations into the role of Bim and other Bcl-2 family members overlapping roles in branching morphogenesis, due to their ability to impact not only levels of apoptosis and proliferation but the ECM milieu as well as cell adhesion and migration.
Reviewer 3 Report
The authors failed to answer my concerns.
Author Response
We are sorry we failed to address your specific concerns. Respectfully, it was not our intent to claim that the molecular mechanisms were identical between ureteric bud and retinal capillary branching. Rather we intended to draw parallels to similar roles Bim plays in branching morphogenesis. As mentioned in response to reviewer 2, Bim has both apoptotic and non-apoptotic roles during development. We feel this protein has overlapping roles in both retinal vascular and ureteric bud development by pruning of unnecessary branches for example. Given the role expression of Bim plays in modulation of ECM protein expression as well as cell adhesion and migration, its expression could impact both branching and pruning. Given the prominent role apoptosis plays during development we believe it is not too farfetched that Bim expression could have some similar impact in multiple organs.